# Parent reports of children's emotional and behavioral problems in a low- and middle-income country (LMIC): An epidemiological study of Nepali schoolchildren

Jasmine Ma[1,2]*, Pashupati Mahat[3], Per Håkan Brøndbo[4], Bjørn H. Handegård[1], Siv Kvernmo[5], Anne Cecilie Javo[1,6]

1 Regional Centre for Child and Youth Mental Health and Child Welfare -North, Faculty of Health Sciences, UiT The Arctic University of Norway, Tromsø, Norway, 2 Child & Adolescent Psychiatry Clinic, Kanti Children's Hospital, Kathmandu, Nepal, 3 Centre for Mental Health and Counseling, Kathmandu, Nepal, 4 Department of Psychology, Faculty of Health Sciences, UiT The Arctic University of Norway, Tromsø, Norway, 5 Department of Clinical Medicine, Faculty of Health Sciences, UiT The Arctic University of Norway, Tromsø, Norway, 6 Sami National Competence Center for Mental Health, Sami Klinihkka, Finnmark Hospital Trust, Karasjok, Norway

* jasminema2006@yahoo.com

**Data Availability Statement:** We have now submitted de-identified data set in this revised submission. An anonymized SPSS data set

## Abstract

### Background

As epidemiological data on child mental health in low- and middle-income countries are limited, a large-scale survey was undertaken to estimate the prevalence and amount of child emotional and behavioral problems (EBP) in Nepal as reported by the parents.

### Methods

3820 schoolchildren aged 6–18 years were selected from 16 districts of the three geographical regions of Nepal, including rural, semi-urban and urban areas. We used the Nepali version of the Child Behavior Checklist (CBCL)/6-18 years as screening instrument. Comparisons of child problems between genders and between the seven largest castes and ethnic groups were carried out by analysis of variance. Prevalence was computed based on American norms.

### Results

Adjusted prevalence of Total Problems was 18.3% (boys: 19.1%; girls:17.6%). The prevalence of internalizing problems was higher than externalizing problems. The mean scores of Total, Externalizing, and Internalizing problems were 29.7 (SD 25.6), 7.7 (SD 8.0), and 9.1 (SD 8.1), respectively. The Khas Kaami (Dalit) group scored the highest, and the indigenous Tharu group scored the lowest on all scales. In the Mountains and Middle Hills regions, problem scores were higher in the rural areas, whereas in the Tarai region, they were higher in the urban areas.

necessary to replicate our study findings, uploaded as a "Supporting information file".

**Funding:** Our study is funded by Child Workers in Nepal (CWIN) / Solidarity Action for Development, Norway FORUT. The funders had no role in the study design, data collection and analysis, decision to publish, or preparation of the manuscript.

**Competing interests:** The authors have declared that no competing interests exist.

## Conclusion

The prevalence and magnitude of emotional and behavioral problems in Nepali children were found to be high compared to findings in meta-analyses worldwide. Problem scores varied according to gender, castes /ethnic groups, and living areas. Our findings highlight the need for a stronger focus on child mental problems in a low-and middle-income country like Nepal.

## Introduction

One third of the world's population are children, with the vast majority living in low- and middle-income countries (LMIC) [1]. Many mental disorders start during childhood and adolescence [2, 3]. Early psychiatric disorders may have a huge effect on children's lives and on the functioning of their families [4]. Several studies have pointed to early identification and treatment of these disorders as key factors for improved prognosis [5]. However, in many LMICs, mental health conditions and disabilities in children have largely gone unrecognized, and early interventions and appropriate service designs for child mental health problems are lacking. [3, 6, 7]. Epidemiological studies may enable the assessment of service needs in LMICs as well as the identification of high-risk groups needing special attention.

A recent review on the global coverage of prevalence data for mental disorders in children aged 5–17 years reported that most of the LMICs had no data on any kind of mental disorders. It further reported that many LMICs were poorly represented in the available data; for example, no region in sub-Saharan Africa had more than 2% coverage for any disorder [8]. Although sparse, previous research in LMICs suggests that child and adolescent mental health problems are common. A systematic review in non-referred samples from LMICs showed a prevalence of about 10–20% in most of the 16 surveys, which is consistent with findings from high income countries (HIC) [3].

A former meta-analysis of 51 Asian countries reported a general prevalence of 10–20% [9]. In the South Asian countries, studies on child mental health are sparse and their quality varies. In India, a review study involving both school-based and community-based studies, reported a prevalence of 23.3% in the school-based studies and 6.5% in the community-based studies. The discrepancy was probably due to several methodological and sample factors in the latter studies [10]. In a systematic review from Bangladesh, the prevalence of mental disorders was found to range from 13.4% to 22.9% among children aged 2–16 [11]. In Pakistan, a study among school children aged 6–16 reported a prevalence of 15.9% of behavior problems and 22.5% of emotional problems [12]. In another LMIC country, Iran, a community-based study of children aged 6–17 reported a prevalence rate of 16.7% of total difficulties [13]. In China, a recent, large-scale epidemiological study in the Sichuan Province reported an overall prevalence of 19.1% of child mental disorders [14].

For Nepal, we do not know the prevalence or magnitude of emotional and behavioral problems (EBP) in the child population as no larger studies have been published internationally. However, an unpublished Nepali PhD dissertation from 2007 on teacher-reported problems of children aged 6–18, using the Children's Behavior Questionnaire (CBQ) as screening instrument, suggested a prevalence rate of 24.5% [15]. Recently, a pilot study of a national mental health survey for Nepal was carried out which included adults and adolescents aged 13–17 years [16]. Using a diagnostic instrument (Mini International Neuropsychiatric Interview-MINI), the prevalence of mental disorders in this age group was found to be 11.2%. However,

broader and more robust survey studies of EBP in Nepali child population comprising several age groups are warranted in order to fill the knowledge gap [17]. The present study is the first large-scale survey of child EBP in a broader age group (6–18 years) that is published internationally.

Although epidemiological studies have consistently identified different types of emotional and behavioral problems for boys and girls [18] there has been little research examining gender differences in child EBP in LMICs, including Nepal [19]. Most of the studies done in Nepal suggest higher rates of behavior problems in boys than in girls with boys having more externalizing problems and girls having more internalizing problems [15, 20, 21]. However, a broader, nationwide sample comprising children of all age groups is needed to confirm these findings.

It should be noted that in many LMICs, the population is multi-ethnic. Internationally, cross-cultural studies have shown that different cultural contexts might play a role in the prevalence and types of child mental problems as culture both defines and creates specific sources of distress [3, 22]. In cross-cultural studies, parents' interpretation and rating of child problems have been shown to differ across cultures, resulting in differences in the prevalence of child mental health problems [23, 24]. A large, cross-cultural study from 45 societies nested within 10 culture clusters reported that society plus culture cluster accounted for about 10% of the variance in parents' ratings of children's problems [25]. Although other factors than culture may play a more important role in ratings of child EBP than society and culture, culture's influence on parents' ratings of child problems are still important to investigate, particularly for the planning of child psychiatric services and for clinical interventions, especially so in the less investigated LMICs. In the present study, we therefore decided to explore possible within-country cultural differences by comparing EBP between the seven largest castes and ethnic groups.

As Nepal is a highly heterogeneous country not only culturally but also when it comes to geography/ ecology and types of living areas, we decided to compare the magnitude of EBP between the main geographic/ecological regions of the country and between the different types of settlements / living areas (rural, semi-urban and urban). By including within-country diversity in our study, we were able to capture a more nuanced picture of the distribution of EBP in the general Nepali child population. Living conditions in the geographic regions of Nepal differ, being harsher in the Mountains region [26], which might influence child mental health. Hence, assessing and comparing the magnitude child EBP between the three main regions of Nepal might be of interest to the authorities in their planning of mental health services. Internationally, several studies have demonstrated that types of living areas might have an impact on child mental health problems [20, 27, 28], but till date, no nationwide study in Nepal has examined and compared child EBP between different types of living areas.

The specific aims of the present study were to assess the prevalence and magnitude (mean scores) of parent reported EBP in Nepali school children aged 6–18. To examine within-country diversity, we compared child problems between a) castes/ethnic groups, b) geographic regions, and c) types of living area. Finally, we looked for gender differences in EBP.

## Materials and methods

### Study site and population

Nepal has a population of 29.6 million people (2021) and is topographically divided into three regions: The Himalaya (Mountain Region) to the North, the Middle Hills region, and the Tarai (the Southern flatland). There are 16 districts in the Mountain Region, 39 districts in the Middle Hills and 20 districts in the Tarai region. According to the Nepali demographic-social census [26], children below 18 years of age represent 44.4% of the total population: 22.5% boys

and 21.9% girls. There are 126 different castes and ethnic groups. The term "caste" basically refers to a group of people who follows Hinduism. Traditionally, Hindu castes are ranked hierarchically in the following order of social status: (1) Brahmins (highest class), (2) Chhetri, (3) Vaishya, and (4) Sudra, also called Dalit (lowest class). The other ethnic groups in Nepal are indigenous nationalities and tribes, known collectively as the Janajati / Adivasi group. They have their own traditional cultures and specific languages, and do not necessarily adhere to, or fall under the Hindu caste system. According to the census [26], the seven largest castes and ethnic groups are: Chhetri 16.6%, followed by Brahmin-Hill 12.2%, Magar 7.1%, Tharu 6.6%, Tamang 5.8%, Newar 5%, and Khas Kaami (the largest group of Dalits) 4.8% [26]. The Magar, Tharu, Tamang, and Newar all belong to the Janajati/Adivasi indigenous group.

## Study design

The present study is a cross-sectional, cross-cultural epidemiological study in the general child population of Nepal.

## Subjects and procedure

**Sampling method.** Based on the population distribution of the three main ecological/geographic regions of Nepal (i.e. 8% of the total population in the Mountain region, 45% in the Middle Hills region, and 48% in the Tarai region), we purposively selected three districts from the Mountain region, and six districts each from the Middle Hills and the Tarai regions. As our study includes an examination of child problems in different castes and ethnic groups, we wanted to ensure a high number of participants in each of these groups. Hence, Kathmandu district was added to the sample because of its multi-cultural population. In all, 16 districts were purposively selected from all over the country. Next, we purposively selected four schools in each district (two government schools and two private schools) based on accessibility and feasibility–i.e. a total of 64 schools in the 16 districts. Our study is a large, countrywide study and required an extensive amount of time and money to accomplish. The purposive sampling technique was chosen for cost effectiveness and for ease of data collection and travels. Students from grades 1 to 10 with six students in each grade (three boys and three girls), were then randomly selected using random number tables. Thus, in each district, 240 children were selected, which gave a total of 3840 children.

**Procedure.** All schoolchildren aged 6–18 were eligible for this general population study, irrespective of their caste and ethnic background. In Nepal, children from all castes, religions and ethnic groups are admitted to the regular schools. Hence, recruiting parents through the regular school system would provide a reasonable cross-section of the child population. Only regular schools were included (i.e. both governmental and private schools), whereas the very few special education schools for children with severe disabilities and faith-based schools representing minor, more segregated groups were excluded. Children's caste and ethnic belongings were classified according to their parents' own definition.

Twenty research assistants (RA) with a bachelor's degree in education / psychology collected the data, and seven supervisors with a master's degree in education or in psychology and experience in data collection work supervised them. Before commencing their work, they all followed an intensive three days' training program administered and led by the researcher (first author) which included orientation about the research project and the instruments, their own role and responsibilities in the project, and a thorough training in how to inform parents and teachers, how to answer queries that might arise, and how to assist in filling in the forms of illiterate parents. Data collection work was monitored by the researcher (first author) by means of frequent telephone check-ups, SKYPE meetings, and by direct visits to the different

districts. A meeting with the school management was conducted at each school and a written consent was obtained. An invitation letter was then sent to the parents, and both oral and written information was provided. Informed consent was obtained from all participating parents. Only mothers were used as informants. Fathers were not included due to capacity problems. For illiterate parents, the research assistants verbally posed the questions to them and helped fill in the forms. Data were collected during September 2017 –January 2018. Plotting of data was done manually during the first half of 2018 by three research assistants, supervised and monitored by the researcher. The overall participation rate was 99.5%. The proportion of missing items was not more than 0.1% for any of the CBCL items.

**Measures.** Based on separate focus group conversations among teachers, parents and professionals about which screening tool would be the most appropriate for Nepal: the Achenbach System of Empirically Based Assessment (ASEBA) / Child Behavior Checklist (CBCL)/6-18 [29] or the Strength and Difficulties Questionnaire (SDQ 4–17) [30], we concluded that the ASEBA instruments would be the better instruments to use (i.e. for detecting problems, for clear and simple questions, and for higher cultural sensitivity).

*1. Child Behavior Checklist (CBCL/6-18).* The CBCL has been translated to more than 100 languages and has established good psychometric properties across the world, including good criterion-related validity, good test-retest reliability, and good internal consistency as measured by Cronbach's alpha [29]. It consists of 20 competence items and 113 problem items. The problem items are scored on eight syndrome scales, two broad-band subscales: Internalizing and Externalizing, and a Total Problems scale. The syndrome scales: Withdrawn/ Depressed, Somatic Complaints and Anxious/Depressed together form the "Internalizing" scale and the scales: Rule-breaking Behavior and Aggressive Behavior together form the "Externalizing" scale. The Social Problems, Attention Problems and Thought Problems scales do not belong to either subscales but are included in the Total Problems scale, which is derived by summing up the individual item scores. The response format of questions on behaviors is: 0 = not true, 1 = somewhat or sometimes true, and 2 = very true or often true.

We used the Nepali version of the Child Behavior Checklist (CBCL)/6-18 that had been translated into Nepali in connection with a former Nepali study [15]. The teacher version (TRF) and the youth version (YSR) of the ASEBA instruments had both been validated and found acceptable for use in Nepal as reported in other studies [15, 20], whereas the parent version (CBCL) had not been validated in Nepali studies before. In our study, we found an acceptable internal consistency for the parent version (CBCL) as indicated by Cronbach's alphas for the eight syndrome scales: Withdrawn / Depressed: 0.71; Somatic Complaints: 0.79; Anxious / Depressed: 0.76; Rule-breaking Behavior: 0.76; Aggressive Behavior: 0.88; Social Problems: 0.73; Attention Problems: 0.80; Thought Problems: 0.75.

*2. Background information questionnaire.* The parents were asked to fill in a questionnaire asking about various background information data. In the present paper, we present the following selected variables: child gender, caste / ethnicity of the child, ecological / geographic region, types of living area, types of school, parents' occupation, and parents' educational level.

**Statistical analyses.** The ASEBA data management and SPSS statistics version 22.0 for Windows were used for all analyses. When computing the overall prevalence for Nepal, sampling weights were used to account for the oversampling for some regions and age-groups (i.e. according to the Nepal Census, 2011) [26]. We used Pearson's chi square test for comparisons between groups on categorical variables. To assess group differences for continuous variables, analysis of variance (ANOVA) was done. For group comparisons involving more than three groups, post hoc comparisons were made using the Scheffé method. To indicate effect size, Hedges' g was computed when comparing two groups. Partial eta squared was the selected effect size when more than two groups were compared. The significance level used in all tests was 0.005.

### Ethical considerations and confidentiality of data

Before commencing the study, ethical approval was obtained from the Ethical Review Board of Nepal Health Research Council (NHRC) (ref. no. 1875; reg, no: 71/2017). Both collection and storage of data were done according to their rules. The records from the study were kept strictly confidential and locked down so that no persons other than the researcher had access to them. All electronic information was coded and secured using a password protected file, and all personally identifiable information was removed from the data set in order to protect the participants' individual privacy. No information will be shared or published that would make it possible to identify any participant.

## Results

### Background data

In Table 1, selected demographic background data are presented for the seven largest castes and ethnic groups (N = 3148), omitting the "Others" group (N = 672). As can be seen from the table, boys and girls were almost equally distributed between the different groups. Most participants, irrespective of caste or ethnic belonging, lived in semi-urban areas. Parents from the Tharu and Khas Kaami (Dalit) groups were the most illiterate, whereas the Newar group had the highest educational level. A substantial number of parents were migrant workers.

### Prevalence of EBP for boys and girls–Total sample

In Table 2, we have presented prevalence as to the Achenbach classification of "normal", "borderline" and "clinical" status according to American norms, both for the Total Problems scale and for the Externalizing and the Internalizing scales. Approximately, one fifth of all children had problems in the clinical range. The prevalence of internalizing problems was higher than externalizing problems.

### Adjusted prevalence for Nepal

Since the Mountain region was somewhat over-sampled and the Middle Hills and Tarai regions under-sampled for 6-18-years-olds in our study, we computed sampling weights that took population numbers in the child population among the three geographic regions into consideration as well as the age distribution, both based on the Nepali 2011 census [26]. As a result, the prevalence of CBCL Total Problems in Nepali 6-18-year-olds who scored in the clinical range was estimated to 18.3%; boys: 19.1% and girls: 17.6%.

### Prevalence of EBP between the different castes and ethnic groups

The prevalence of child EBP varied among the different castes and ethnic groups (Table 3). It was highest for the Khas Kaami (Dalit) group and lowest for the indigenous Tharu group.

### The magnitude of EBP for boys and girls–Total sample

Table 4 shows mean scores for the whole sample by gender. Boys scored significantly higher than girls on Total Problems, Externalizing Problems, as well as on the three subscales: Social Problems, Thought Problems and Attention Problems. However, there were no gender differences in mean scores for the Internalizing scale. The effect sizes for the gender comparisons can be considered as small according to Cohen (1988) [31], with Hedges' g ranging from 0.02 to 0.20.

**Table 1. Distribution of selected demographic variables for the seven largest castes and ethnic groups.**

| Castes and ethnic groups | Chhetri | Hill Brahmin | Magar | Tharu | Tamang | Newar | Khas Kaami (Hill Dalit) |
|---|---|---|---|---|---|---|---|
| N (%) | 866 (22.7) | 905 (23.7) | 187 (4.9) | 246 (6.4) | 335 (8.8) | 162 (4.2) | 447 (11.7) |
| **Background variables** | | | | | | | |
| *Gender* | | | | | | | |
| Boys | 427 (49.3) | 457 (50.5) | 93 (49.7) | 124 (50.4) | 180 (53.7) | 68 (42.0) | 235 (52.6) |
| Girls | 439 (50.7) | 448 (49.5) | 94 (50.3) | 122 (49.6) | 155 (46.3) | 94 (58.0) | 212 (47.4) |
| *Geographic Location* | | | | | | | |
| Mountain | 125 (14.4) | 158 (17.5) | 11 (5.9) | 1 (0.4) | 121 (36.1) | 6 (3.7) | 49 (11.0) |
| Hill | 431 (49.8) | 457 (50.5) | 128 (68.4) | 27 (11.0) | 116 (34.6) | 111 (68.5) | 265 (59.3) |
| Tarai | 310 (35.8) | 290 (32.0) | 48 (25.7) | 218 (88.6) | 98 (29.3) | 45 (27.8) | 133 (29.8) |
| *Rural/Semi-urban/Urban[1]* | | | | | | | |
| Rural | 227 (26.2) | 242 (26.7) | 28 (15.0) | 19 (7.7) | 111 (33.1) | 19 (11.7) | 124 (27.7) |
| Semi-urban | 423 (48.8) | 489 (54.0) | 117 (62.6) | 200 (81.3) | 181 (54.0) | 74 (45.7) | 259 (57.9) |
| Urban | 216 (24.9) | 174 (19.2) | 42 (22.5) | 27 (11.0) | 43 (12.8) | 69 (42.6) | 64 (14.3) |
| *Types of School* | | | | | | | |
| Governmental | 359 (41.5) | 321 (35.5) | 91 (48.7) | 171 (69.5) | 195 (58.2) | 68 (42.0) | 348 (77.9) |
| Private | 507 (58.5) | 584 (64.5) | 96 (51.3) | 75 (30.5) | 140 (41.8) | 94 (58.0) | 99 (22.1) |
| *Mother's Occupation* | | | | | | | |
| Housewife | 596 (68.8) | 606 (67.0) | 134 (71.7) | 192 (78.0) | 237 (70.7) | 100 (61.7) | 319 (71.4) |
| Public Service | 53 (6.1) | 59 (6.5) | 6 (3.2) | 5 (2.0) | 14 (4.2) | 4 (2.5) | 10 (2.2) |
| Private Business | 69 (8.0) | 102 (11.3) | 20 (10.7) | 13 (5.3) | 24 (7.2) | 29 (17.9) | 36 (8.1) |
| Farmer | 116 (13.4) | 91 (10.1) | 16 (8.6) | 26 (10.6) | 39 (11.6) | 17 (10.5) | 58 (13.0) |
| Migrant Worker | 7 (0.8) | 13 (1.4) | 2 (1.1) | 3 (1.2) | 9 (2.7) | 2 (1.2) | 11 (2.5) |
| Others | 25 (2.9) | 34 (3.8) | 9 (4.8) | 7 (2.8) | 12 (3.6) | 10 (6.2) | 13 (2.9) |
| *Father's Occupation* | | | | | | | |
| Private business | 192 (22.2) | 266 (29.4) | 26 (13.9) | 54 (22.0) | 58 (17.3) | 68 (42.0) | 76 (17.0) |
| Farmer | 264 (30.5) | 251 (27.7) | 46 (24.6) | 88 (35.8) | 118 (35.2) | 31 (19.1) | 163 (36.5) |
| Migrant worker | 150 (17.3) | 131 (14.5) | 45 (24.1) | 28 (11.4) | 65 (19.4) | 20 (12.3) | 88 (19.7) |
| Others | 126 (14.5) | 120 (13.3) | 25 (13.4) | 65 (26.4) | 71 (21.2) | 28 (17.3) | 94 (21.0) |
| *Family Education[2]* | | | | | | | |
| Illiterate | 83 (9.6) | 44 (4.9) | 10 (5.3) | 57 (23.2) | 47 (14.0) | 8 (4.9) | 76 (17.0) |
| Primary Level[3] | 154 (17.8) | 147 (16.2) | 51 (27.3) | 71 (28.9) | 137 (40.9) | 26 (16.0) | 175 (39.1) |
| Secondary Level[4] | 528 (61.0) | 544 (60.1) | 117 (62.6) | 110 (44.7) | 141 (42.1) | 83 (51.2) | 182 (40.7) |
| University Level[5] | 101 (11.7) | 170 (18.8) | 9 (4.8) | 8 (3.3) | 10 (3.0) | 45 (27.8) | 14 (3.1) |

[1]The place of residence (rural, semi-urban, urban) was defined according to the official classifications made by the Ministry of Federal Affairs & General Administration (MOFAGA) and further verified by parent's own reports.

[2]In the households with two parents, the higher education level was used.

[3] Primary level of education consists of grade 1 to 8.

[4] Secondary level of education consists of grade 9 to 12.

[5] University level includes Bachelor, Masters or PhD degree.

## The magnitude of EBP in the different castes and ethnic groups

Table 5 presents the comparison of mean scores on the different problem scales between the seven largest castes and ethnic groups using one-way ANOVA. The Khas Kaami (Dalit) group scored the highest and the Tharu group scored the lowest on all scales. In the post hoc multiple comparisons, the Tharu group differed from the Chhetri, the Brahmin Hill and the Khas Kaami groups on Total Problems, Externalizing Problems, Social Problems, Thought

**Table 2. Prevalence of EBP for boys and girls–Total sample.**

| | Gender | | Total (N = 3820) |
|---|---|---|---|
| | *Male* | *Female* | |
| | *(N = 1914)* | *(N = 1906)* | |
| *Total Problems T score* | | | |
| Normal (<60) | 68.7% | 71.5% | 70.1% |
| Borderline (60–63) | 11.2% | 10.4% | 10.8% |
| Clinical (>63) | 20.1% | 18.1% | 19.1% |
| *Internalizing problems T score* ** | | | |
| Normal (<60) | 61.9% | 66.9% | 64.4% |
| Borderline (60–63) | 12.7% | 10.3% | 11.5% |
| Clinical (>63) | 25.4% | 22.8% | 24.1% |
| *Externalizing problems T score* * | | | |
| Normal (<60) | 76.5% | 80.3% | 78.4% |
| Borderline (60–63) | 7.9% | 6.8% | 7.4% |
| Clinical (>63) | 15.6% | 12.9% | 14.2% |

*$P<0.05$;

**$P<0.005$;

***$P<0.0005$. For gender comparisons, the Pearson's chi square test was used.

Problems and Attention Problems. For Internalizing Problems, the Tharu group differed only from the Chhetri and Khas Kaami, whereas the Tamang group differed from the Khas Kaami and Chhetri groups. However, the effect sizes were small, with partial eta squared ranging between 0.010 to 0.015.

## Associations between EBP and geographic region and types of area

There were no differences in the magnitude of problems between the three geographic regions, except for higher Internalizing Problems in the Mountain region. However, there were

**Table 3. Prevalence of EBP for the seven largest castes and ethnic groups.**

| | Chhetri | Brahmin- Hill | Magar | Tharu | Tamang | Newar | Khas Kaami | Total |
|---|---|---|---|---|---|---|---|---|
| *Total Problems Tscore* *** | | | | | | | | |
| Normal (<60) | 65.6% | 71.8% | 73.3% | 78.9% | 72.5% | 70.4% | 63.3% | 69.5% |
| Borderline (60–63) | 12.1% | 9.5% | 8.6% | 8.1% | 13.1% | 13.6% | 9.6% | 10.7% |
| Clinical (>63) | 22.3% | 18.7% | 18.2% | 13.0% | 14.3% | 16.0% | 27.1% | 19.8% |
| *Internalizing Problems Tscore* *** | | | | | | | | |
| Normal (<60) | 60.5% | 64.5% | 67.4% | 73.6% | 67.5% | 66.7% | 58.4% | 63.8% |
| Borderline (60–63) | 12.0% | 11.6% | 12.3% | 8.5% | 12.5% | 10.5% | 8.7% | 11.2% |
| Clinical (>63) | 27.5% | 23.9% | 20.3% | 17.9% | 20.0% | 22.8% | 32.9% | 25.0% |
| *Externalizing Problems Tscore* *** | | | | | | | | |
| Normal (<60) | 76.2% | 77.3% | 81.3% | 88.2% | 82.4% | 78.4% | 72.0% | 77.9% |
| Borderline (60–63) | 6.6% | 8.4% | 7.0% | 4.9% | 7.2% | 8.0% | 6.9% | 7.2% |
| Clinical (>63) | 17.2% | 14.3% | 11.8% | 6.9% | 10.4% | 13.6% | 21.0% | 14.9% |

*$P<0.05$;

**$P<0.005$;

***$P<0.0005$. For group comparisons, the Pearson's chi square test was used.

**Table 4. The magnitude of EBP for boys and girls–Total sample.**

|  | Boys (N = 1914) Mean (SD) | Girls (N = 1906) Mean (SD) | Total (N = 3820) Mean (SD) | Gender effect F | Effect size g [a] |
|---|---|---|---|---|---|
| Total Problems | 31.19 (26.67) | 28.14 (24.47) | 29.67 (25.64) | 13.54 *** | 0.11 |
| Externalizing Problems | 8.44 (8.52) | 6.86 (7.35) | 7.65 (7.99) | 37.35 *** | 0.19 |
| Internalizing Problems | 9.01 (8.09) | 9.21 (7.96) | 9.11 (8.03) | 0.57 | -0.02 |
| Social Problems | 3.50 (3.25) | 3.18 (3.02) | 3.34 (3.14) | 9.92 ** | 0.10 |
| Thought Problems | 2.44 (3.07) | 2.11 (2.79) | 2.27 (2.94) | 12.17 *** | 0.11 |
| Attention Problems | 4.24 (3.75) | 3.52 (3.36) | 3.88 (3.58) | 38.62 *** | 0.20 |

*$P<0.05$;

**$P<0.005$;

***$P<0.0005$;

[a]Hedges' g.

significant interactions between geographic regions and types of living area on Total Problems, as well as on the two broadband scales (Table 6). In the Mountain and Middle Hills regions, the problem scale scores were higher in the rural areas than in the semi-urban or urban areas. In contrast, children living in rural areas in the Tarai region scored lower than those living in semi-urban and urban areas. The sizes of the interaction effects were small, with partial eta squares less than 0.01.

## Discussion

This study assessed the prevalence and amount of EBP among schoolchildren in Nepal and compared the prevalence and magnitude of problems among different groups based on gender, caste / ethnicity, and types of area (urban, semi-urban and rural).

### Multicultural norms of the CBCL

Based on data from 31 societies, Achenbach and Rescorla constructed different norm groups (high, medium, and low) for the CBCL, based on the omni-cultural mean of 22.5 (SD 5.6) that was found by averaging the Total Problem scores of the 31 cultures [32]. Nepal has not yet been included in the ranking of countries due to the lack of internationally published scientific studies. However, the present study, showing a Total Problems mean score of 29.7 (SD 25.6), suggests that Nepal should be placed under the high scoring countries.

**Table 5. The magnitude of EBP in the different castes and ethnic groups.**

|  | Chhetri N = 866 Mean (SD) | Brahmin-Hill N = 905 Mean (SD) | Magar N = 187 Mean (SD) | Tharu N = 246 Mean (SD) | Tamang N = 335 Mean (SD) | Newar N = 162 Mean (SD) | Khas Kaami N = 447 Mean (SD) | Group effect F | Partial Eta squared |
|---|---|---|---|---|---|---|---|---|---|
| Total problems | 32.67 (27.55) | 29.95 (25.23) | 27.16 (25.65) | 22.53 (23.59) | 26.76 (22.21) | 28.82 (22.38) | 34.15 (28.70) | 8.15*** | 0.015 |
| Externalizing Problems | 8.24 (8.55) | 7.78 (7.91) | 6.48 (7.08) | 5.69 (7.07) | 6.69 (7.36) | 7.82 (7.39) | 8.89 (8.77) | 6.04*** | 0.011 |
| Internalizing Problems | 10.18 (8.87) | 9.15 (7.78) | 8.48 (7.98) | 7.19 (7.50) | 8.04 (6.67) | 8.77 (7.25) | 10.40 (9.08) | 7.59*** | 0.014 |
| Social Problems | 3.71 (3.32) | 3.38 (3.06) | 3.14 (3.35) | 2.50 (2.77) | 3.02 (2.99) | 3.10 (2.71) | 3.77 (3.53) | 6.95*** | 0.013 |
| Thought Problems | 2.54 (3.09) | 2.24 (2.92) | 2.20 (3.04) | 1.58 (2.60) | 1.97 (2.44) | 2.12 (2.36) | 2.80 (3.47) | 6.38*** | 0.012 |
| Attention Problems | 4.22 (3.75) | 3.92 (3.62) | 3.77 (3.70) | 3.01 (3.18) | 3.66 (3.55) | 3.74 (3.32) | 4.38 (3.73) | 5.11*** | 0.010 |

*$P<0.05$;

**$P<0.005$;

***$P<0.000$.

**Table 6. CBCL scores by geographic regions and types of living area- Total sample.**

|  | Total Problems | Internalizing Problems | Externalizing Problems |
|---|---|---|---|
|  | Mean (SD) | Mean (SD) | Mean (SD) |
| **Mountain** |  |  |  |
| Rural Area (N = 352) | 37.08(30.20) | 12.03(9.92) | 9.17(8.94) |
| Semi-Urban Area (N = 134) | 22.82(21.30) | 7.60(6.91) | 5.19(6.23) |
| Urban Area (N = 0) | - | - | - |
| Total (N = 486) | 33.15(28.73) | 10.81(9.39) | 8.07(8.46) |
| **Hills** |  |  |  |
| Rural Area (N = 460) | 32.92(28.12) | 9.87(8.68) | 8.61(8.99) |
| Semi-Urban Area (N = 902) | 29.46(23.73) | 9.14(7.62) | 7.62(7.34) |
| Urban Area (N = 556) | 28.60(23.19) | 8.68(7.41) | 7.31(7.28) |
| Total (N = 1918) | 30.04(24.75) | 9.18(7.84) | 7.77(7.76) |
| **Tarai** |  |  |  |
| Rural Area (N = 58) | 20.05(20.99) | 5.69(6.62) | 5.55(6.22) |
| Semi-Urban Area (N = 1117) | 26.93(25.83) | 8.27(7.72) | 7.08(8.10) |
| Urban Area (N = 241) | 34.69(25.83) | 9.90(7.57) | 9.12(8.54) |
| Total(N = 1416) | 27.97(25.59) | 8.44(7.69) | 7.36(8.15) |
| Main effect size of Geographic Region | F = 2.86 | F = 6.81** | F = 1.12 |
|  | $\eta^2 = 0.002$ | $\eta^2 = 0.004$ | $\eta^2 = 0.001$ |
| Main effect size of Rural, Semi-Urban and Urban Areas | F = 14.54*** | F = 8.64*** | F = 10.63*** |
|  | $\eta^2 = 0.008$ | $\eta^2 = 0.005$ | $\eta^2 = 0.006$ |
| Effect of Interaction between Geographic Regions and Rural, Semi-Urban and Urban Areas | F = 13.85*** | F = 12.94*** | F = 10.014*** |
|  | $\eta^2 = 0.011$ | $\eta^2 = 0.010$ | $\eta^2 = 0.008$ |

*$P<0.05$;

**$P<0.005$;

***$P<0.0005$; $\eta^2$partial eta square.

## Comparison of results with other studies

We found that the percentage of Nepali children who scored in the clinical range, i.e. above the American cut-off, was 19.1% for Total Problems, with an adjusted prevalence of 18.3%. Compared to the overall prevalence of mental health problems for schoolchildren in Asian countries as reported in an earlier review [9], the prevalence for Nepal seems high. However, the prevalence is consistent with findings from school studies in neighboring countries, e.g. China: 19.1% [14] and India: 23.3% [10].

The relatively high prevalence of child problems in these and other LMICs might be due to a higher level of environmental risk factors such as natural disasters [33, 34] and adverse social circumstances like poverty and child abuse and neglect [35, 36]. Social disadvantages and family fragmentation (e.g. caused by migrant working parents) are known to elevate level of stress affecting the mental health of parents as well as children [37, 38]. A possible explanation for the higher problem scores in Nepal could be the exposure to the devastating earthquake that hit the country in 2015 and the traumatic events that followed in its wake. The mental health effect of exposure to disasters like earthquakes, especially on children, is still largely unknown, and various resilience and posttraumatic growth (PTG) factors may be involved [39]. However, the present study was not designed to address a possible link between child EBP and disaster exposure, and this hypothesis should be considered with caution. Other studies comparing areas hit by the earthquake with areas not affected are needed to confirm it. Further, the higher

prevalence of EBP may be due to the poor and stressful living conditions experienced by many families in a LMIC country like Nepal. Yet another reason might be that Nepali parents might have a different threshold for reporting child EBP due to cultural norms. Cultural differences affecting parents' ratings and interpretation of child behavior have been explored to a rather small extent internationally, and to our knowledge, no such studies have been performed in Nepal. Finally, it should be noted that the higher prevalence might be due to methodological reasons. In our study, we used a screening instrument (CBCL) and studies using screening instruments may yield higher prevalence rates than studies using diagnostic tools [9].

We found a higher level of Total Problems and Externalizing Problems in boys than in girls. This finding converges with findings from other international studies [18]. It also converges with the earlier Nepali dissertation study by Mahat mentioned in the introduction [15], suggesting the same gender pattern for Nepal. However, we did not find more Internalizing Problems in girls than in boys, contrasting to the findings from international meta-studies [18, 40]. Our finding may be due to cultural or methodological factors and needs replication for verification. More studies on gender patterns in child EBP are warranted across cultures, especially from the less investigated LMICs.

An interesting finding in our study was the higher prevalence of internalizing problems (24.1%) compared to externalizing problems (14.2%). The finding converges with a recent epidemiological study from Kenya [41]. Like in the Kenyan study, a possible explanation for the elevated Internalizing Problems score could be the higher awareness and subsequent higher scorings of somatic symptoms by the parents. In Nepal, where there is very little awareness of mental problems in general, and particularly in children, parents tend to pay more attention to their child's physical symptoms than their conduct and may define problems accordingly. Besides, Nepali children, like children from other South Asian countries, are socialized to control their frustrations and negative emotions, i.e. to internalize their problems, rather than acting them out [42]. Internationally, cross-cultural studies have shown significant variations in the relative dominance of internalizing versus externalizing problems [40, 43]. Due to different cultural norms and different socio-religious contexts, the types of problems that children express will differ. In countries where the culture discourages child aggression and other uncontrolled behaviors, internalizing problems like shyness, anxiety, and depression are noted more often, whereas in cultures that accept acting-out of emotions, externalizing behaviors are noted more often [40].

## Within-country differences in EBP

Cross-cultural comparisons showed that the Khas Kaami (Dalit) group had the highest prevalence of EBP. In Nepal, experiences of caste-based discrimination are found to be prevalent among the Dalit [44] and may be one of the main reasons for the higher problem level. In contrast, the Tharu, who mostly live in small villages in the Tarai region, showed the lowest amount of EBP. One possible explanation for the lower scores in this tribal, indigenous group may be that it has some strong protective factors, both at the individual, family, and social level. However, a more detailed investigation among the Tharu people exploring family and social factors that may influence child EBP is warranted. Further, parent ratings may differ across ethnic groups depending upon differences in cultural norms and attitudes of reporting problems. Internationally, studies have shown that parents of ethnic minorities may be less likely to perceive problem behavior in their children when compared to ethnic majority parents [45, 46]. A third explanation may be that the linguistic problems as well as the high level of illiteracy among the Tharu parents might have interfered with their ratings in such a way that reporting problems became more difficult. The Tharu people have their own mother language which differ from the majority Nepali language. Lack of language skills might have

hampered the communication between the parents and the research assistants as well as making the perception of questions more demanding.

We observed only small differences in the amount of child problems between geographic regions. However, some interesting interactions emerged for regions with types of living area. In the Mountain and Hills regions, children who lived in the rural areas had the highest problem scores, whereas in the Tarai region, children in the urban areas scored the highest. The higher problem scores in the rural areas in the Mountain region may be due to poverty and the tougher living conditions that exist in those areas [26]. The higher amount of problems found in the urban areas of the Tarai region may be due to the migration of families from the countryside to the cities in search of better working opportunities. Urbanization might lead to adjustment problems, more stress exposure, and increased vulnerability due to factors like overcrowding, low social support, inadequate security, and increased violence [47, 48]. The disparities in our findings converge with findings from the international literature. Some studies have found more child problems in rural areas [27], whereas others have found more problems in urban areas [47]. As countries differ in their economic development and cultural orientations, rural—urban differences in one country may not be generalized to other countries [28]. Based on the present study, we argue that this may be the case within a specific country as well–especially in countries as diverse as Nepal.

## Strengths and limitations of the study

This epidemiological study is the first internationally published study on the prevalence and amount of EBP in Nepali schoolchildren as reported by their parents. We have used one of the most wide-spread and internationally validated instruments to assess child problems: the CBCL/6-18. The Cronbach's alpha in our study was above 0.7 for all eight syndrome scales. We used sound methodology and thorough procedures in data collection, including helping illiterate parents with the filling in of forms and reaching out to parents who could not manage to visit the school. This probably increased parents' trust and willingness to participate in the study and resulted in a very high participation rate: 99.5%.

There are some methodological limitations of the study. First, our prevalence estimation was done by using the American norms of the CBCL as Nepali norms do not yet exist. Hopefully, future studies may provide separate norms for Nepal. Although we collected our data in all the main geographic regions of Nepal and in 16 districts in different parts of the country, we cannot claim that the results are representative for the whole country. Further, the number of participants in some of the castes and ethnic groups were small, which might have affected the results and made them less reliable. Hence, future studies with larger samples are recommended for those groups to confirm the findings.

Another limitation is that fathers' reports were not assessed. Generally, fathers who are substantively involved in their children's lives, may provide valuable information about their children's problems [49].

The present paper focuses on the magnitude of child problems and how problems may vary across gender, different cultural groups, and types of living area. It does not include an examination of family- and social correlates of EBP. Such associations will be the focus in a subsequent paper currently in progress, and with the same sample of Nepali children.

## Conclusion

The study provides new knowledge about the prevalence of child EBP in a LMIC. It amply demonstrates that in a country like Nepal, many children may suffer from various types of mental problems which may need attention. Furthermore, it highlights the importance of

taking into account possible gender- and cultural differences in the magnitude and types of child problems, as well as pointing to rural–urban differences. The findings may be useful to the health authorities in developing child- and adolescent mental health services. Finally, the study provides important background information for both clinicians and teachers in dealing with child mental health problems.

## Supporting information

**S1 File.**
(SAV)

## Acknowledgments

We are grateful to all participating parents and schools, and the team of data enumerators and supervisors for making this study possible. Further, we would like to thank Dr. Arun Raj Kunwar and his child and adolescent psychiatry team at Kanti Children's Hospital, Kathmandu, for their support.

## Author Contributions

**Conceptualization:** Jasmine Ma, Anne Cecilie Javo.

**Formal analysis:** Jasmine Ma.

**Methodology:** Jasmine Ma, Anne Cecilie Javo.

**Supervision:** Pashupati Mahat, Per Håkan Brøndbo, Bjørn H. Handegård, Siv Kvernmo, Anne Cecilie Javo.

**Writing – original draft:** Jasmine Ma.

**Writing – review & editing:** Pashupati Mahat, Per Håkan Brøndbo, Bjørn H. Handegård, Siv Kvernmo, Anne Cecilie Javo.

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
