## [Decision Letter · Decision Letter 0]

27 Apr 2021

PONE-D-21-07867

Parent reports of child behavior problems in a low- and middle- income country (LMIC): An epidemiological study of Nepali school children

PLOS ONE

Dear Dr. Ma,

Thank you for submitting your manuscript to PLOS ONE. After careful consideration, we feel that it has merit but does not fully meet PLOS ONE’s publication criteria as it currently stands. Therefore, we invite you to submit a revised version of the manuscript that addresses the points raised during the review process.

We look forward to receiving your revised manuscript.

Kind regards,

Pranil Man Singh Pradhan

Academic Editor

PLOS ONE

Additional Editor Comments:

1.Title: The title mentions the outcome as child behavior problems. The aims of the study mentions the outcome of the study as EBP. It is advised to maintain consistency in the terminology used specially in the title and objectives.

2. Abstract line 32: Replace means with mean score

3. Line 70: Please expand the acronym EBP upon first use.

4. Introduction: As it stands the introduction mentions mostly about the prevalence of child mental disorders worldwide. It would be better to also mention about the significance of child mental disorders in terms of clinical as well as public health impact.

Line 94: Aims of the study - Separate heading is not required. It can be merged with the last paragraph of introduction.

Line 137: Please justify in the manuscript why children with special needs/faith based schools were excluded.

Line 99: Materials and methods: It will be better to include a separate section for ethical considerations. Also mention how the privacy and confidentiality of the participants was maintained.

Line 169-170: Please clarify whether the CBCL tool was administered to the parent or the child. Were both teacher and youth versions used in this study? Also were both parents interviewed or a single parent?

Line 191: Please confirm whether the significance level was set at 0.05 or 0.005.

Line 238: Better term for amount would be magnitude.

Journal Requirements:

Reviewers' comments:

Reviewer's Responses to Questions

**Comments to the Author**

1. Is the manuscript technically sound, and do the data support the conclusions?

Reviewer #1: Yes

Reviewer #2: Yes

2. Has the statistical analysis been performed appropriately and rigorously? 

Reviewer #1: Yes

Reviewer #2: I Don't Know

3. Have the authors made all data underlying the findings in their manuscript fully available?

Reviewer #1: Yes

Reviewer #2: Yes

4. Is the manuscript presented in an intelligible fashion and written in standard English?

Reviewer #1: Yes

Reviewer #2: Yes

5. Review Comments to the Author

Reviewer #1: 1.Why did authors decided on taking caste and ethnicity based differences in behavioural problems in children? A literature support may be needed at the introduction part.

2. The term "lowest Hindu case" may be ethically wrong to use. It may be used suggesting it to be the prevalent in Nepal but the authors may refrain from using term "lowest or highest" caste in the results and discussion part.

3. Is there any support of ecology and mental health as authors have tried to see that in three ecological zones of Nepal?

4. As purposive sampling technique was used what purpose of the sample collection was taken into account (ease of data collection, ease of travel, ease of …)

5. What does "referral from school" mean in the methodology? Please clarify

6. Why were faith based schools excluded? Also what happened to the Christian, Muslim or Buddhist children? Were they excluded? And the Muslims, Christians or Buddhists may not follow the caste based classification, hence limiting the horizons of this research.

7. How was training of research assistant done? At one point authors mentioned "Plotting ……

, supervised and controlled by the researcher". What does controlled here mean?

8. Table 2 and 3 mention about the "p value" in the footnote of tables but what statistical test was done and what was compared? Please clarify.

9. The classification of "urban, semi urban and rural" how was this done. Please provide reference to this classification.

10. Regarding the discussion authors point out that earthquake could be a reason for high behavioural problems. This should be mentioned with caution as not whole nation had its impacts. It should also be seen what sample areas taken in this study were hit by earthquake?

11. Was the ascent taken from the children in study?

12. Suggestions:

a. Please see if the journal allows discussion with separate head in each finding. If yes I am ok

b. Please cite the following landmark studies in the introduction from Nepal as we have data on prevalence

• Jha, A. K., Ojha, S. P., Dahal, S., Sharma, P., Pant, S. B., Labh, S., Marahatta, K., Shakya, S., Adhikari, R. P., Joshi, D., Luitel, N. P., & Dhimal, M. (2019). Prevalence of Mental Disorders in Nepal: Findings from the Pilot Study. Journal of Nepal Health Research Council, 17(2), 141-147. https://doi.org/10.33314/jnhrc.v0i0.1960

• http://nhrc.gov.np/wp-content/uploads/2020/09/Factsheet-Adolescents.pdf

Reviewer #2: Congratulations for such a wonderful study from LMIC like Nepal covering all geographical areas.

The study seems to have been well conducted.

There are some minor grammatical errors that need correction which have been mentioned in the attached word file itself.

Here are few major questions that need to be addressed:

1. The Khas Kaami (Hill Dalit) has been disproportionately represented in this study, in comparison to their population size, and the same participants have been found to have more problems. So, the reason for this disproportionate representation needs further clarification.

2. The total number of participants and that of boys and girls is different in different tables which have been mentioned in the attached comments as well. This needs correction/clarification.

3. The literacy status of the parents of the participants of Tharu ethnic group seems to be much low as compared to that of others and their reporting of child EBP is also the lowest. Some discussion into this aspect might be worthy.

4. The composition of family structure- nuclear vs joint/extended greatly affect the child rearing and thus could affect the EBP of the children as well. It would be worthy to discuss if the authors have collected data regarding this part.

5. In line 375 "Another limitation is that fathers’ reports were not assessed." has been mentioned. No such thing has been mentioned anywhere in the methods section. Does it mean that data were collected from "mothers" only. If so, it needs to be mentioned explicitly in the methods section as well.

6. PLOS authors have the option to publish the peer review history of their article (what does this mean?). If published, this will include your full peer review and any attached files.

Reviewer #1: No

Reviewer #2: **Yes: **Madhur Basnet, MD(Psychiatry), Associate Professor, Department of Psychiatry, B. P. Koirala Institute of Health Sciences, Dharan, Nepal

---

## [Author Response · Author response to Decision Letter 0]

18 Jun 2021

Response to reviewers

Thank you all for your valuable suggestions as to revision of the manuscript. Below is our response. We have tried to give you our answers to the questions posed one by one and in doing so, we have referred to the particular lines in our revised manuscript with track changes where we have put the revisions (see Revised manuscript with track changes). 

As for grammatical errors and language improvements / clarifications, we have noted that they should be addressed at revision as the journal does not copyedit accepted manuscripts. Hence, we have made some minor corrections and clarifications in the text: line 1; line 25; line 43; line 77; line 80; line 141; line 156-157; line 162; 173; line 200; line 223 – 225; line 237; line 260; line 330; line 439; line 443. 

Editorial Manager: 

1. In the revised manuscript attached to the editorial manager mail sent 12.04.21, you asked us to use terms consistently, and to use the term “Nepali”, not “Nepalese”. This has now been done (line 19 and 40).

2. Further, you asked us to use the full form while using abbreviation for the first time, such as EBP (emotional and behavior problems). This has now been corrected (line 77–78).

3. Thank you for pointing to an error made in the section about sampling method. We have now corrected it by changing the wording: “the capital city of Kathmandu” to “Kathmandu district” (line 160). 

4. Table 1 had an error (missing number) in one of the rubrics (0 instead of 10). Thank you for noticing. This has now been corrected (line 264). 

5. You commented that there were no ** and *** in the Table 2. Unfortunately, one star was missing for the Internalizing problems T score. It should have been two stars, not one as the gender difference was significant at the 0.005 level. We have now corrected it in the revised manuscript (line 278). In the footnote below the table, we explained the meaning of the different stars: *P<0.05; **P<0.005; ***P<0.0005. Although we used all of them in the tables to indicate different levels of significance, in the text itself we have only commented on results that had a significant level **P<0.005 or lower (i.e. not *P<0.05) as we defined the significance level for this study as 0.005 (as stated in the “Statistical analyses” section). 

6. You asked why the problem scores in the clinical range in Table 3 and Table 2 differed. The reason for this is that the numbers of informants were not the same. In Table 2, we estimated the prevalence for the total sample (N=3820), whereas in Table 3, we estimated the prevalence for the seven largest castes and ethnic groups (N=3148) while omitting the “Others” group (N=672). We have tried to make this clearer and more precise in the revised manuscript by adding the number (N) for the total sample in Table 2 (line 278), and we have changed the heading of Table 1 and 3 from “…. the different castes and ethnic groups” to “… the seven largest castes and ethnic groups”, thus implying that this did not include the whole sample (line 264 and 292). The numbers (N) for the different castes and ethnic groups are already presented in Table 1. However, we made the preceding text to the Table 1 clearer by adding information about the total number for all seven groups (N=3148), and also added that we had omitted the “Others” group (N=672) - i.e. informants who belonged to other cultural groups (line 257 – 258). 

7. You asked why there was a difference in the number of boys and girls and total sample between Table 1 and Table 4 as the total sample in Table 1 was 3148 while in Table 4 it was 3820. The answer is the same: Table 1 included the seven largest groups only (N=3148), whereas Table 4 included the total sample (N=3820). 

Additional Editor:

Comment 1: Title: The title mentions the outcome as child behavior problems. The aims of the study mention the outcome of the study as EBP. It is advised to maintain consistency in the terminology used specially in the title and objectives.

Our reply: Thank you for noticing. We have now revised the title of the paper so that it includes both emotional and behavioral problems. EBP is maintained consistently throughout the whole manuscript.

Comment 2: Abstract line 32: Replace means with mean score

Our reply: Thank you for your suggestion. We have now replaced the word “mean” with “mean score” (line 33) and whereever applicable.

Comment 3: Line 70: Please expand the acronym EBP upon first use.

Our reply: Thank you for rightly pointing it out. We have now done so (line 77-78).

Comment 4: Introduction: As it stands, the introduction mentions mostly about the prevalence of child mental disorders worldwide. It would be better to also mention about the significance of child mental disorders in terms of clinical as well as public health impact.

Line 94: Aims of the study - Separate heading is not required. It can be merged with the last paragraph of introduction.

Our reply: We have revised the introduction accordingly by writing more about the impact of child mental disorders, the importance of early identification for improved prognosis, and the need for early interventions and appropriate service designs in LMICs, linking it to the importance of conducting epidemiological studies (line 46 – 56). 

We have now removed the separate heading for Aims of the study and have put this paragraph as the last paragraph of the introduction (line 128). 

Comment 5: Line 137: Please justify in the manuscript why children with special needs/faith based schools were excluded.

Our reply: Our study is a study of EBP in the general child population of Nepal. We therefore recruited parents through the regular schools (i.e. both governmental and private schools). In Nepal, children from all faiths, castes, and ethnic groups, including Muslims, Christians and Buddhists, are admitted to the regular schools. Hence, we believed that regular schools would provide a reasonable cross-section of the Nepali child population. In our own sample, both Muslim, Christian and Buddhist children were represented. Based on this argument, and also because we wanted to compare child EBP between the seven largest castes and ethnic groups as defined by the Nepali census and not between the different religions, we excluded special faith-based schools. Besides, to include faith-based schools might have led to overrepresentation of certain groups. Further, we excluded special education schools for children with severe disabilities because such severe conditions along with the special environment that these schools represent, might grossly influence EBP. In the revised manuscript, we have clarified the nature of our study (i.e. a study in the general child population) and explained that regular schools in Nepal admit children of all faiths, castes and ethnicities so that recruiting parents through the regular school system would provide a reasonable cross-section of the child population. We also added that special schools in Nepal are very few compared with regular schools (line 173-179). 

Comment 6: Line 99: Materials and methods: It will be better to include a separate section for ethical considerations. Also mention how the privacy and confidentiality of the participants was maintained.

Our reply: We have now included a separate section for ethical considerations as the last paragraph of “Materials and methods”. Here, we have described how the privacy and confidentiality of the participants were maintained (line 246 – 254). 

Comment 7: Line 169-170: Please clarify whether the CBCL tool was administered to the parent or the child. Were both teacher and youth versions used in this study? Also were both parents interviewed or a single parent?

Our reply: The CBCL is the parent version of the ASEBA instruments and was administered to the parents. The Teacher report form (TRF), i.e. the teacher version of the ASEBA, and the Youth self-report (YSR), i.e. the youth version of the ASEBA, were not used in the present study. In this study, only mothers filled in the forms, not the fathers. In the revised manuscript, we have informed that only mothers were used as informants and that fathers were not included due to capacity problems (line 194-195). 

Comment 8: Line 191: Please confirm whether the significance level was set at 0.05 or 0.005.

Our reply: The significance level was set at 0.005 as described in the Statistical Analyses section: “The significance level used in all tests was 0.005” (line 244-245). See also our reply to the Editor Manager, point 5, about the use of stars in the tables to mark different significance levels. 

Comment 9: Line 238: Better term for amount would be magnitude.

Our reply: Thank you for your suggestion. We have revised the manuscript and changed the term “amount of problems” to “magnitude of problems” (line 295) and wherever it occurred. 

Reviewer #1: 

Comment 1: Why did authors decided on taking caste and ethnicity based differences in behavioural problems in children? A literature support may be needed at the introduction part.

Our reply: We have revised the Introduction part so that it now includes a literature support for cross-cultural comparisons of child EBP (line 97-112). We have referred to international studies that have demonstrated cross-cultural differences in child problems across countries and ethnic groups, suggesting that such differences might exist between castes and ethnic groups in Nepal as well. 

Comment 2: The term "lowest Hindu case" may be ethically wrong to use. It may be used suggesting it to be the prevalent in Nepal but the authors may refrain from using term "lowest or highest" caste in the results and discussion part.

Our reply: Thank you for your suggestion. In our study, we found the highest prevalence of EBP in the Khas Kaami group which is the largest Dalit group and the term “lowest caste” was used to indicate that this group is an under-privileged group that may experience caste-based discrimination. According to your advice, we have now deleted the term. Instead, we have used “Dalit” in parenthesis (line 35; 261; 290; 307; 393) which we believe is better known than “Khas Kaami” among international readers. However, we have kept the information about the Hindu hierarchical caste system in the “Study site and population” part.

Comment 3: Is there any support of ecology and mental health as authors have tried to see that in three ecological zones of Nepal?

Our reply: We did not have any theoretically based reason for looking at possible differences in child EBP in the three main geographical / ecological regions of Nepal. Our intent was to explore the magnitude of child problems in the different regions of the country in order to capture a more nuanced picture of the distribution of EBP in Nepal. We thought that perhaps there might be more child EBP in the Mountain region compared to the other two regions due to the harsher living conditions. We believed that more knowledge about possible differences in the geographic distribution of problems might be of interest to the Nepali health authorities and useful in future child mental health programs. We have now explained more thoroughly why we decided to explore possible geographical differences in the Introduction (line 113 – 127). 

Comment 4: As purposive sampling technique was used what purpose of the sample collection was taken into account (ease of data collection, ease of travel, ease of …)

Our reply: Purposive sampling technique was used for the selection of districts and schools and was based on accessibility and feasibility. However, the selection of the students was based on a random sampling technique. Our study is a large, countrywide study and required an extensive amount of time and money to accomplish. A purposive sampling technique was chosen for cost effectiveness and for ease of data collection and travels. The explanation is now given in the revised manuscript (line 162 – 167). 

Comment 5: What does "referral from school" mean in the methodology? Please clarify

Our reply: We agree that this wording is confusing. The sentence is now corrected in the revised manuscript, and the word “feasibility” is used instead (line 163). The whole sentence now reads as follows: “Next, we purposively selected four schools in each district (two government schools and two private schools) based on accessibility and feasibility - i.e. a total of 64 schools in the 16 districts».

Comment 6: Why were faith based schools excluded? Also what happened to the Christian, Muslim or Buddhist children? Were they excluded? And the Muslims, Christians or Buddhists may not follow the caste based classification, hence limiting the horizons of this research.

Our reply: Our study is a survey in the general child population of Nepal. As children from all faiths, castes, and ethnic groups, including Muslims, Christians and Buddhists, are admitted to the regular schools, we chose to recruit parents through the regular school system. We believed that regular schools would provide a reasonable cross-section of the Nepali child population. In our own sample, both Muslim, Christian, and Buddhist children were represented. There are few faith-based schools in the country, representing only a small percentage of the total number of schools. By including them, some religious groups might have been overrepresented in our sample. Further, this study intended to compare children from the seven largest castes and ethnic groups in Nepal, as defined by the Nepali census, and these children are all attending regular schools. Hence, we decided to exclude faith-based schools. We have now explained about this in the revised paper (line 173-179).

Comment 7: How was training of research assistant done? At one point authors mentioned "Plotting ……, supervised and controlled by the researcher". What does controlled here mean?

Our reply: In the revised manuscript, we have now described in more detail how the training of the research assistants was done (page 182 – 190). 

We agree that the word “controlled” is not a correct word. We have now changed it into “monitored” which is the correct word to use (line 199). 

Comment 8: Table 2 and 3 mention about the "p value" in the footnote of tables but what statistical test was done and what was compared? Please clarify.

Our reply: As mentioned in the Statistical Analysis paragraph, we used Pearson’s chi square test for comparisons between groups on categorical variables. In the revised manuscript, we have now added what statistical test we used in Table 2 and 3 and for what comparisons. This is put as footnotes below the tables (line 279-280 and line 293-294). 

Comment 9: The classification of "urban, semi urban and rural" how was this done. Please provide reference to this classification.

Our reply: The place of residence / municipality (i.e. rural, semi-urban, and urban) was defined according to the official classifications made by the Ministry of Local Development (GoN) and further verified by parents’ own reports. This has now been informed in the revised manuscript as a footnote below Table 1 (line 265-266). 

Comment 10: Regarding the discussion authors point out that earthquake could be a reason for high behavioural problems. This should be mentioned with caution as not whole nation had its impacts. It should also be seen what sample areas taken in this study were hit by earthquake?

Our reply: Thank you for your suggestion. As described in the manuscript, the aim of our study was not to demonstrate the impact of the 2015 earthquake, and we did not design the study so that it could demonstrate any connection between the earlier disaster and the magnitude of child EBP. We just mentioned the earthquake to suggest that we might possibly find a higher prevalence of child mental problems due to this disaster’s possible prolonged effect on mental health. We also mentioned that there might be many other factors, such as poverty and other social / family problems, that might have an impact on the level of child mental problems in Nepal. According to your advice, we have now added in the revised manuscript that this hypothesis should be considered with caution and that other studies comparing areas hit by the earthquake with areas not affected, are needed in order to confirm it (line 357 – 359). 

Comment 11: Was the ascent taken from the children in study?

Our reply: The consent was taken from the parents only. This was approved by the ethical approval board of NHRC (approval granted in 2017).

Comment 12: Suggestions:

a. Please see if the journal allows discussion with separate head in each finding. If yes I am ok

Our reply: We have now deleted the following subheadings in the Discussion part: Prevalence (line 341); Gender differences (line 368); Types of problems (line 376); Castes and ethnic groups (line 393); Regions and types of living areas (line 411). 

b. Please cite the following landmark studies in the introduction from Nepal as we have data on prevalence

• Jha, A. K., Ojha, S. P., Dahal, S., Sharma, P., Pant, S. B., Labh, S., Marahatta, K., Shakya, S., Adhikari, R. P., Joshi, D., Luitel, N. P., & Dhimal, M. (2019). Prevalence of Mental Disorders in Nepal: Findings from the Pilot Study. Journal of Nepal Health Research Council, 17(2), 141-147. https://doi.org/10.33314/jnhrc.v0i0.1960

• http://nhrc.gov.np/wp-content/uploads/2020/09/Factsheet-Adolescents.pdf

Our reply: Thank you for the suggestion. We have now cited the above-mentioned article from Nepal by Jha et al (2019) in the introduction and presented the prevalence of mental disorders that was found for the age group 13-17 (line 81-84). 

Reviewer #2: 

Congratulations for such a wonderful study from LMIC like Nepal covering all geographical areas. The study seems to have been well conducted. There are some minor grammatical errors that need correction which have been mentioned in the attached word file itself.

Here are few major questions that need to be addressed:

Comment 1: The Khas Kaami (Hill Dalit) has been disproportionately represented in this study, in comparison to their population size, and the same participants have been found to have more problems. So, the reason for this disproportionate representation needs further clarification.

Our reply: The Khas Kaami group represents the seventh of the largest castes and ethnic groups in Nepal according to the Nepali census of 2011, i.e. 4.8% of the total population. In our study, they constitute 11.7% of the total sample. The reason for this disproportionate representation may be the selection of districts or / and schools. Some of the districts / schools included in the study may have had a larger Khas Kaami child population which then explain the higher number of Khas Kaami children. Also, the randomly selection of students from each school might by chance have provided a higher number of Khas Kaami children. However, this incongruity is not decisive as to the comparisons of mean scores of EBP between groups as the statistical tests used in this study are able to handle differences in group sizes. 

Comment 2: The total number of participants and that of boys and girls is different in different tables which have been mentioned in the attached comments as well. This needs correction/clarification.

Our reply: We have explained about this in our reply to the Editor Manager. Please, see our answer to point 6 in his list of comments. The point is that the numbers of informants were not the same in Table 2 and 3. In Table 2, we estimated the prevalence for the total sample (N=3820), whereas in Table 3, we estimated the prevalence for the seven largest castes and ethnic groups (N=3148) while omitting the “Others” group (N=672). We have clarified this in the manuscript (see line numbers in our reply to the Editor Manager).

Comment 3: The literacy status of the parents of the participants of Tharu ethnic group seems to be much low as compared to that of others and their reporting of child EBP is also the lowest. Some discussion into this aspect might be worthy.

Our reply: Thank you for your suggestion. The lower literacy status of the Tharu parents found in our study (illiteracy 23.2%) is in line with the data obtained by the national census of Nepal (2011) stating that the educational status of the Tharus was poor and that the percentage of illiteracy in the Tharu population was 36.1%. In the Discussion section of the revised manuscript, we have now discussed whether illiteracy and language problems might have interfered with the Tharu parents’ rating of problems (line 404 – 410).

Comment 4: The composition of family structure- nuclear vs joint/extended greatly affect the child rearing and thus could affect the EBP of the children as well. It would be worthy to discuss if the authors have collected data regarding this part.

Our reply: The aims of the present paper were to examine the prevalence and magnitude of child EBP in the different regions and living areas of Nepal, and to compare possible gender and ethnic differences. However, social and family correlates of child problems, such as family structure, was not part of the present study. As mentioned in the manuscript, exploring such correlations will be the focus of our next paper, now in progress (line 447 – 450).

Comment 5: In line 375 "Another limitation is that fathers’ reports were not assessed." has been mentioned. No such thing has been mentioned anywhere in the methods section. Does it mean that data were collected from "mothers" only. If so, it needs to be mentioned explicitly in the methods section as well.

Our reply: Thank you for this reminder. We have now added that mothers only were used as informants and that fathers were not included due to capacity problems. We have put this information in the Methods part (line 194 – 196).

---

## [Decision Letter · Decision Letter 1]

21 Jul 2021

Parent reports of children’s emotional and behavioral problems in a low- and middle- income country (LMIC): An epidemiological study of Nepali schoolchildren

PONE-D-21-07867R1

Dear Dr. Ma,

We’re pleased to inform you that your manuscript has been judged scientifically suitable for publication and will be formally accepted for publication once it meets all outstanding technical requirements.

Kind regards,

Pranil Man Singh Pradhan

Academic Editor

PLOS ONE

Additional Editor Comments (optional):

Reviewers' comments:

Reviewer's Responses to Questions

**Comments to the Author**

1. If the authors have adequately addressed your comments raised in a previous round of review and you feel that this manuscript is now acceptable for publication, you may indicate that here to bypass the “Comments to the Author” section, enter your conflict of interest statement in the “Confidential to Editor” section, and submit your "Accept" recommendation.

Reviewer #1: All comments have been addressed

Reviewer #2: All comments have been addressed

2. Is the manuscript technically sound, and do the data support the conclusions?

Reviewer #1: Yes

Reviewer #2: Yes

3. Has the statistical analysis been performed appropriately and rigorously? 

Reviewer #1: Yes

Reviewer #2: I Don't Know

4. Have the authors made all data underlying the findings in their manuscript fully available?

Reviewer #1: Yes

Reviewer #2: Yes

5. Is the manuscript presented in an intelligible fashion and written in standard English?

Reviewer #1: Yes

Reviewer #2: Yes

6. Review Comments to the Author

Reviewer #1: (No Response)

Reviewer #2: Thank you for addressing all the comments well. The manuscript is now much refined, clearer and better.

7. PLOS authors have the option to publish the peer review history of their article (what does this mean?). If published, this will include your full peer review and any attached files.

Reviewer #1: **Yes: **Dr. Pawan Sharma

Reviewer #2: No

---

## [Editor Report · Acceptance letter]

26 Jul 2021

PONE-D-21-07867R1 

Parent reports of children’s emotional and behavioral problems in a low- and middle- income country (LMIC): An epidemiological study of Nepali schoolchildren 

Dear Dr. Ma:

I'm pleased to inform you that your manuscript has been deemed suitable for publication in PLOS ONE. Congratulations! Your manuscript is now with our production department. 

Kind regards, 

on behalf of

Dr. Pranil Man Singh Pradhan 

Academic Editor

PLOS ONE